# Haptoglobin-Related Protein without Signal Peptide as Biomarker of Renal Salt Wasting in Hyponatremia, Hyponatremia-Related Diseases and as New Syndrome in Alzheimer’s Disease

**DOI:** 10.3390/biom13040638

**Published:** 2023-04-01

**Authors:** John K. Maesaka, Louis J. Imbriano, Candace Grant, Nobuyuki Miyawaki

**Affiliations:** Department of Medicine, Division of Nephrology and Hypertension, NYU Langone Hospital Long Island and NYU Long Island School of Medicines, Mineola, New York, NY 11501, USA

**Keywords:** haptoglobin-related protein without signal peptide, cerebral salt wasting, biomarker of renal salt wasting, renal salt wasting in Alzheimer’s disease

## Abstract

The application of pathophysiologic tenets has created significant changes in our approach to hyponatremia and hyponatremia-related conditions. This new approach incorporated the determination of fractional excretion (FE) of urate before and after the correction of hyponatremia and the response to isotonic saline infusion to differentiate the syndrome of inappropriate secretion of antidiuretic hormone (SIADH) from renal salt wasting (RSW). FEurate simplified the identification of the different causes of hyponatremia, especially the diagnosis of a reset osmostat and Addison’s disease. Differentiating SIADH from RSW has been extremely difficult because both syndromes present with identical clinical parameters, which could be overcome by successfully carrying out the difficult protocol of this new approach. A study of 62 hyponatremic patients from the general medical wards of the hospital identified 17 (27%) to have SIADH, 19 (31%) with reset osmostat, and 24 (38%) with RSW with 21 of these RSW patients presenting without clinical evidence of cerebral disease to warrant changing the nomenclature from cerebral to renal salt wasting. The natriuretic activity found in the plasma of 21 and 18 patients with neurosurgical and Alzheimer’s disease, respectively, was later identified as haptoglobin-related protein without signal peptide (HPRWSP). The high prevalence of RSW creates a therapeutic dilemma of deciding whether to water-restrict water-logged patients with SIADH as compared to administering saline to volume-depleted patients with RSW. Future studies will hopefully achieve the following: 1. Abandon the ineffective volume approach; 2. Develop HPRWSP as a biomarker to identify hyponatremic and a projected large number of normonatremic patients at risk of developing RSW, including Alzheimer’s disease; 3. Facilitate differentiating SIADH from RSW on the first encounter and improve clinical outcomes.

## 1. Introduction

Recent studies in hyponatremia and hyponatremia-related conditions have significantly altered our diagnostic and therapeutic approaches that need to be updated in a review and developed in the future. These changes were made possible by abandoning the inaccurate categorization of hyponatremia that is based on the status of the extracellular volume. This approach has been in existence for over 50 years despite a general awareness that we cannot accurately determine the volume status by clinical criteria in non-edematous patients [1]. The new approach utilized well-established pathophysiologic concepts that identified the different causes of hyponatremia with greater accuracy. The most difficult and important clinical application of this approach was to differentiate the syndrome of inappropriate secretion of antidiuretic hormone (SIADH) from cerebral, or more appropriately renal salt wasting (RSW). The difficulty in differentiating SIADH from RSW can be explained in part by the commonly perceived rarity of RSW and the uncommon realization that both syndromes are characterized by identical clinical parameters (Table 1).

Table listing identical clinical and laboratory findings in SIADH except for their volume status. All parameters are used clinically to characterize both diseases. One key difference that can-not be assessed clinically is the volume status, being increased in SIADH and decreased in RSW.

Both SIADH and RSW present with hyponatremia, normal renal, thyroid and adrenal function, hypouricemia, concentrated urine defined as urine osmolality (Uosm) being greater than the plasma osmolality (Posm), urine sodium concentration (UNa) usually exceeding 30 mmol/L and high fractional excretion (FE) of urate. Due to the divergent therapeutic goals of water-restricting water-logged patients with SIADH and administering salt and water to volume-depleted patients with RSW, it is essential to differentiate between these two syndromes with identical clinical parameters. This differentiation would be unnecessary if RSW was indeed a rare condition. The identification of patients with RSW, which included patients without clinical evidence of cerebral disease, suggested that a study of hyponatremic patients from the general medical wards of the hospital would test our increasing awareness that RSW might be more common than perceived [2,3,4]. The pathophysiologic studies, which differentiated SIADH from RSW, were difficult to execute and were only possible by the presence of a research coordinator in a funded study. Some studies conducted in the general medical wards increased our awareness that RSW was more common than perceived, including patients without clinical evidence of cerebral disease to warrant a change in nomenclature from cerebral to RSW [2,4,5].

This study unexpectedly found 24 (38%) of 62 hyponatremic patients recruited from the general medical wards of the hospital to have RSW. The absence of cerebral disease in 21 of these 24 patients supported our proposal to change cerebral to RSW to justify this important change in the nomenclature [2,4,5]. These studies underscored the need to re-evaluate the manuscripts that made the diagnosis of SIADH by simply meeting the overlapping parameters without going through the rigors of differentiating one syndrome from the other. In this review, we will describe the development of a more pathophysiologic approach to hyponatremia and hyponatremia-related conditions that appears to have identified the different causes with greater clarity and finally we will review the studies leading to identification of HPRWSP as the natriuretic factor that causes RSW. Eventual development of HPRWSP as a biomarker of RSW will simplify the diagnosis of RSW on the first encounter and improve clinical outcomes [6].

Our recent identification of the natriuretic protein, haptoglobin-related protein without signal peptide (HPRWSP), found in the plasma of salt wasting patients with neurosurgical and Alzheimer’s diseases, has the potential of serving as a biomarker to simplify the identification of RSW patients on the first encounter to improve clinical outcomes [6].

## 2. New Pathophysiologic Approach to Hyponatremia and Hyponatremia-Related Conditions

The recent report of a high prevalence of RSW in the general medical wards of the hospital by the application of well-developed pathophysiologic principles provided sufficient data of high credibility to abandon the ineffective volume approach. It took 30 years to generate consistently reproducible data to propose this new approach. There are two pathophysiologic phenomena that eventually led to the identification of HPRWSP as the likely natriuretic factor that induces RSW.

### 2.1. Determination of Fractional Excretion (FE) of Urate

Investigations into the unique relationship between serum sodium and FEurate can now reliably differentiate SIADH from RSW. As noted in Figure 1, an increased FEurate of >11% exists in both SIADH and RSW when the patient is hyponatremic. This relationship diverges when FEurate returns to normal in SIADH but remains persistently increased after the correction of the hyponatremia in patients with RSW [2,3,4,7,8,9,10,11,12].

Urate is freely filtered at the glomerulus and is transported exclusively in the proximal tubule of the kidney by reabsorbing and secretory transporters where there is normally a net reabsorption of 89–96% of the filtered load of urate, leaving 4–11% of the filtered load to be excreted in the final urine and designated as FEurate [13]. FEurate can be determined by the following formulae on simultaneously collected blood and urine:$$FE\;urate\;in\;\%=\frac{urine\;urate}{serum\;urate}÷\frac{urine\;creatinine}{serum\;creatinine}\times 100=\frac{urine\;urate\times serum\;creatinine}{urine\;creatinine\times serum\;urate}\times 100\;$$

Utilizing FEurate to identify the different causes of hyponatremia is especially useful in identifying patients with a reset osmostat, which occurs in about 30% of patients with hyponatremia. It is also useful in identifying patients with Addison’s disease. This elusive disease presents with nondescript symptoms such as weight loss, fatigue, nausea, vomiting, postural dizziness and hyperpigmentation where 50% of patients with Addison’s disease are diagnosed at a time of adrenal crisis [14]. The characteristic hyperkalemia of >5.0/mmol/l is present in only 34% of patients as compared to 84% presenting with serum sodium of <137 mmol/l [15]. Mineralocorticoid deficiency, mainly as aldosterone, can induce devastating hypovolemia by decreasing the sodium uptake in the distal tubule. The intact proximal tubule responds to the hypovolemia by increasing solute transport to generate the typical findings of pre-renal azotemia such as an increase in the BUN-to-creatinine ratio and a low FEurate of <4%. Incorporating FEurate in the work-up of hyponatremic patients will yield values that are <4%, which according to the algorithm includes the consideration of Addison’s disease (Figure 1). This algorithm needs to be developed further by including patients with drug-induced hyponatremia, especially those that have an upregulation of the ADH V2 receptor that is characterized by hyponatremia of an SIADH-like picture but with low plasma ADH levels [16].

### 2.2. Response to Isotonic Saline Infusion

Differences in response to an isotonic saline infusion between SIADH and RSW patients can be best explained by the physiologic differences in the mechanisms that increase ADH secretion. ADH levels in SIADH do not respond to changes in the extracellular volume or plasma osmolality and is caused by the autonomous and inappropriate secretion of ADH as compared to a predictive appropriate response to both volume and osmolar stimuli in RSW [4,17]. Infusions of isotonic saline lead to a prompt excretion of sodium and the failure to dilute the urine or correct the hyponatremia, leading to the dictum that the hyponatremia of SIADH is unresponsive to infusions of large volumes of isotonic saline [18].

As noted in Figure 2, the volume stimulus for ADH secretion is more potent than the osmolar stimulus, so the ADH levels in RSW will remain increased as long as the patient remains volume depleted to perpetuate the hypo-osmolality [19]. In this volume-depleted state, the infusion of isotonic saline would remove the more potent volume stimulus to permit the coexisting hypo-osmolality to inhibit the ADH secretion, excrete dilute urines, eliminate the excessively retained water and correct the hyponatremia [2,4,12].

We tested the response to an isotonic saline infusion in patients by making certain we had correctly made the diagnosis of SIADH and RSW. Two patients had SIADH with increased blood volume as determined by radioisotope dilution methods and decreased plasma renin and aldosterone levels and one patient had RSW with decreased blood volume and increased plasma renin and aldosterone levels. As depicted in Figure 3a, an isotonic saline infusion in a patient with SIADH failed to dilute the urine or correct the hyponatremia, which was identical to the second patient with SIADH [2]. In the patient with RSW (Figure 3b), however, an isotonic saline infusion eliminated the volume stimulus for ADH secretion and allowed the coexisting hypo-osmolality to inhibit the ADH secretion to an undetectable level when the urine was dilute 13 h after initiating the isotonic saline infusions and correction of hyponatremia in 48 h [4].

The undetectable ADH level at the time the urine osmolality was dilute at 152 mosm/kg was recently challenged, claiming that the ADH was still present because the urine was not maximally dilute at an expected 50 mosm/kg [20]. This comment is true when applied to a normal subject but incorrect when applied to patients with RSW who are known to have a free water clearing defect because of a high urinary content of sodium or high osmolar clearance. A maximally dilute urine of 50 mosm/kg would be expected when a normal subject ingests large volumes of water without solute. However, when a solution of half-normal saline was infused after accomplishing a minimal Uosm of 50 mosm/kg to maintain the ADH levels at undetectable levels in normal subjects, the infusion of half-normal saline expanded the extracellular volume, increased urine sodium or osmolar clearance and progressively increased urine osmolality that approached plasma osmolality (Figure 4) [21]. An inability to maximally dilute the urine is, therefore, a predictable and well-established physiological outcome of defective free water excretion that defines patients with RSW.

Another example of a failure to apply established physiological phenomena is the study of 49 hyponatremic patients with a subarachnoid hemorrhage [22]. The diagnosis of SIADH was made by simply meeting the criteria for SIADH without considering or going through the difficult task of differentiating it from patients with RSW. In this study, all the patients received an isotonic saline infusion; none of the patients were water-restricted and none received hypertonic saline or an ADH V2 receptor inhibitor. All patients corrected their hyponatremia in a median of 3 days. As depicted in Figure 3a, an isotonic saline infusion does not dilute the urine or correct the hyponatremia in SIADH [2,18]. All 49 hyponatremic patients with a subarachnoid hem

Orrhage actually had RSW, which is an accepted common cause of RSW. Two editorials agreed that all 49 hyponatremic patients had SIADH but failed to acknowledge the well-established physiologic observations that isotonic saline does not correct the hyponatremia in SIADH but typically does correct it in RSW [2,4,7,18,23,24].

## 3. Application of Pathophysiologic Approaches to Evaluating Patients with Hyponatremia

The determination of FEurate and the response to isotonic saline infusions in hyponatremic patients identified an unexpectedly high rate of RSW in hyponatremic patients from the general medical wards of the hospital [7]. Of the 62 patients studied, we found:

Seventeen (27%) had SIADH based on five with baseline high FEurate which normalized after correction of their hyponatremia (Figure 1). Eleven failed to dilute their urines or correct their hyponatremia while receiving ample quantities of isotonic saline.

Nineteen (31%) had a reset osmostat based on a normal FEurate on all patients and eight excreted a diagnostic dilute urine on a spontaneously excreted urine.

Twenty-four (38%) had RSW with eleven demonstrating a persistent increase in FEurate after correcting their hyponatremia. Nineteen diluted their urine within 24 h after initiating isotonic saline infusions with two demonstrating undetectable levels of ADH when their urine was dilute. Ten patients excreting dilute urines required infusions of 5% dextrose in water to prevent correcting serum sodium by more than 6 mmol/L over a 24 h period to reduce the risk of developing osmotic demyelination [7,25]. Of great interest was the absence of clinical evidence of cerebral disease in 21 of the 24 patients with RSW, which provided compelling data to support our previous proposal to change cerebral to RSW [5,7]. This change in the nomenclature is extremely important to execute because RSW would not be considered in the absence of clinically apparent cerebral disease. The perception that RSW is a rare clinical entity plus the absence of cerebral disease would have led to water-restricting these patients for an erroneous diagnosis of SIADH. The increase in morbidity and mortality associated with hyponatremia may, thus, be in part iatrogenic [26], due to Addison’s disease or hydrochlorothiazide.

An additional unexpected finding was the surprisingly low baseline UNa in a significant number of patients. Thirteen patients with RSW had baseline UNa less than 30 mmol/L as compared to three with SIADH and six with a reset osmostat. Because patients with hyponatremia are usually seen when they are in an equilibrated stage of their disease when urine output of sodium matches sodium intake, patients with RSW probably lost their appetite because of more serious comorbid conditions. Determinations of Una, therefore, appear to be less important in our approach to hyponatremic patients, especially when the algorithms do not consider SIADH or even RSW unless their UNa exceeds 30 or 40 mmol/L [27,28].

The study of 62 hyponatremic patient from the general medical wards can be legitimately criticized because of the labor-intensive nature of attempting to determine FEurate before and after correction of their hyponatremia or whether they diluted their urine after initiating isotonic saline infusions. This was only possible because the funded study included a research coordinator whose sole function was to carry out the protocol as precisely as possible, especially when there were so many levels of function that had to be coordinated and at a time when the length of stay was a major issue. The ultimate value of this study was to introduce the importance of considering RSW as a common cause of hyponatremia and to engage those involved in hyponatremia and hyponatremia-related diseases to contribute to a better understanding and improving clinical outcomes for these patients.

## 4. Rat Clearance studies Demonstrating Natriuretic Activity in Plasma from Patients with Neurosurgical and Alzheimer Diseases

As part of our quest to identify the physiologic mechanisms that contribute to clinical diseases, we decided to develop strategies to identify a natriuretic protein that might cause RSW. It became evident that the utilization of an animal model would be most informative because it allowed many competing variables to be expressed as compared to an in vitro model where a limited number of variables are controlled in an in vitro setting. We decided to perform rat clearance studies by infusing plasma from prospective patients with RSW. The most likely candidates were neurosurgical patients where the blood volume studies demonstrated RSW to be very common and in Alzheimer’s disease (AD) where they were hypouricemic, most likely due to an increase in FEurate [29,30]. Because uric acid is exclusively transported in the proximal tubule, we realized that the natriuretic factor must have a dominant effect on solute transport in the proximal tubule, including sodium [13]. We elected to study sodium and lithium transport because lithium it is transported on a one-to-one basis with sodium in the proximal tubule with little or no transport in the distal tubule as compared to sodium, which can be vigorously transported in the distal tubule [31]. Lithium was infused at a constant rate throughout the experiment. In two separate rat clearance studies, we infused the plasma of 21 patients with various types of neurosurgical diseases and 18 patients with advanced AD to improve our chances of demonstrating the presence of a natriuretic factor [32,33]. The infusion of plasma in both studies revealed virtually identical features where there were no changes in blood pressure or glomerular filtration rates in the study groups of rats as compared to the control rats. FEsodium, however, increased from a control of 0.3% and 0.33% to 0.59% and 0.63% in the neurosurgical and Alzheimer’s disease groups, respectively. FElithium increased from a control of 22.3% and 27.2% to 36.6% and 41.7% in the neurosurgical and Alzheimer’s disease groups, respectively [32,33]. Interestingly FElithium in the patients with fairly advanced AD progressively increased from an elevated level at a mini mental status examination (MMSE) score of twelve to zero, Figure 5.

Figure 5 Graph showing the effect of plasma from patients with moderately advanced Alzheimer’s disease and multi-infarct dementia on FElithium in rats. Note how FElithium is unchanged at the different levels of dementia in patients with multi-infarct dementia. In contrast, patients with MMSE below 12, progressively increase FElithium as their dementia worsens to an MMSE of 0. Based on the dose relationship between HPRWSP and FEsodium/FElithium, the blood levels of HPRWSP are increasing as the dementia worsens. Horizontal line represents best fit for FElithium in multi-infarct dementia and oblique line for Alzheimer’s disease.

Because there was a dose response of FElithium to increasing doses of plasma, the blood levels of the natriuretic protein must have been increasing, suggesting that blood levels of the natriuretic factor and dehydration from RSW were increasing as the patient became more demented. It is, thus, likely that all demented patients below an MMSE score of less than 12 were volume depleted from RSW. It is interesting to note that FEurate increased modestly from 6.6 in the control group to 9.7% in the patients with AD [33]. The mean FEurate of 18.7% in the 17 hyponatremic patients with RSW from the general medical wards of the hospital suggest that these AD patients were more volume depleted than those from the general medical wards [7].

## 5. Identification of Natriuretic Protein in Sera of Patients with Neurosurgical and Alzheimer Diseases

Attempts to identify the natriuretic factor(s) that was present in the plasma of patients with neurosurgical and Alzheimer’s disease in 1993 was possible when the analysis of proteins had developed to a point where every protein could be identified with precision in small sample sizes. We modified the protocol of the previous rat renal clearance studies by injecting 0.5 mL of serum in 1.5 min instead of 0.5 mL given intraperitoneally followed 90 min later by a constant infusion of 2 mL administered over a 3 h period [6,32,33]. We found natriuretic activity in the sera of normonatremic patients with evidence of RSW associated with a subarachnoid hemorrhage and another with Alzheimer’s disease. We subjected the sera with natriuretic activity and the control to mass spectrometry, SWATH analysis and subjected the results to RCProtein (fold-change of the protein) to estimate semi-quantitatively the relative change of a specific protein in the active sera as compared to a control serum [6]. Seventeen proteins in the sera with natriuretic activity were increased at least two-fold over the control sample with the highest levels noted for haptoglobins and haptoglobin-related protein (HPR). Recombinant samples of HPR with signal peptide, haptoglobin Hp 1-1, Hp 2-2, kininogen, thrombospondin, PROZ, alpha 1 micryoglobulin/bikunin, and retinol-binding protein had no natriuretic activity. A review of our analytical data revealed that HPR found in the active sera did not possess the signal peptide. The infusion of recombinant HPR without signal peptide (HPRWSP) resulted in a robust dose response increase in FEsodium and urine volume (Figure 6) [6].

Atrial natriuretic peptide (ANP) has been implicated as the natriuretic peptide that causes RSW. The low normal value of the baseline ANP of 35 pg/mL is consistent with the decreased blood volume in the unequivocal case of RSW presented above (Figure 3b) [4]. The infusion of plasma from 21 and 18 RSW patients with neurosurgical diseases and Alzheimer’s disease, respectively, failed to demonstrate the major effect of ANP by not profoundly decreasing blood pressure or increasing glomerular filtration rates in rats [32,33,34,35]. Moreover, ANP had a slight effect on solute transport in the proximal tubule, loop of Henle and collecting duct as compared to a robust effect of HPRWSP in the proximal tubule. ANP also significantly increased potassium excretion as compared to the natriuretic peptide [34]. The increase in sodium excretion appeared to be due to a hemodynamic effect of significantly increasing the glomerular filtration rates with hardly any effect on the tubular transport of solutes [34,35]. The physiologic effects of ANP are, thus, vastly different from what has been demonstrated with the identified peptide and should be eliminated from further discussions about its contribution to RSW.

## 6. Developing HPRWSP as Biomarker for RSW in Hyponatremic and Non-Hyponatremic Patients

It is evident from the foregoing discussion that there is a dire need to develop simpler ways to differentiate SIADH from RSW. It is anticipated that HPRWSP will not be increased in patients with Addison’s disease, Bartter’s or Gittelman’s syndromes. Developing HPRWSP as a biomarker will allow us to differentiate SIADH from RSW on the first encounter with the patients, correctly select the diametrically opposite therapeutic options and improve clinical outcomes. SIADH and RSW occur commonly in hyponatremic patients in the general medical wards of the hospital but there is an untapped potential of identifying RSW in normonatremic patients as well.

## 7. Projected High Prevalence of RSW in Normonatremic Patients

There is an underemphasis of the need to have a sufficiently high water intake to induce hyponatremia. The loss of approximately 500 mL/day of pure water loss by insensible means will induce hypernatremia if there is inadequate water intake. This is especially true of patients with AD who were all normonatremic except for one patient because they did not drink enough water to develop hyponatremia due to a decrease in thirst with aging and their dementia. We should also be aware of the studies of administering daily injections of pitressin to normal subjects and dogs where hyponatremia did not develop without consciously providing sufficient water intake [36,37]. This is also true in all conditions of hyponatremia and is exemplified by the need to limit water intake in patients with SIADH who lose their ability to excrete free water. This is especially pertinent to patients with a subarachnoid hemorrhage, which was a common cause of hyponatremia. Because RSW was considered a rare disease for many years, these patients were misdiagnosed as having SIADH and were water-restricted until water restriction was found to increase morbidity and mortality [38]. It is now common practice to treat these patients with isotonic saline to prevent cerebral ischemia and infarction of the brain if they are fluid restricted, especially if they are volume depleted from RSW. The lack of sufficient water intake in these patients reduces the likelihood of developing hyponatremia and creates a need to identify normonatremic patients with RSW by developing HPRWSP as a biomarker. Demonstration of reduced blood volumes by radioisotope dilution methods in neurosurgical patients without SAH suggests that all neurosurgical patients should be tested for HPRWSP levels in their blood [29,30,32].

As discussed earlier, 18 of 19 patients with advanced AD had progressively increasing FElithium levels as they became more demented or as the MMSE decreased below 12. The dose-dependency of FElithium excretion rates suggests that the blood concentration of HPRWSP was rising as the dementia worsened (Figure 5) [6,33]. Although these patients had significantly higher FEurates than age and gender matched controls and patients with multi-infarct dementia, FEurate had been much higher in other patients with RSW [5]. While isotonic saline is known to have a meagre effect on FEurate in normal subjects, there is a likelihood that FEurate is lowered by a greater degree of volume depletion at the advanced stages of dementia. A significant degree of volume depletion in RSW will lower FEurate as it would in patients at this level of dementia in AD. In the patient with RSW (Figure 3b), the infusion of isotonic saline increased the baseline FEurate of 29.6 to 63% 13 h after initiating the infusion of isotonic saline, suggesting that the volume depletion was more severe in patients with advanced AD [4,33]. Studies are planned to establish HPRWSP as a biomarker for RSW at various levels of dementia in AD and to determine the clinical responses to an inhibitor to HPRWSP that needs to be developed. It should be noted that HPRWSP, a relatively unknown peptide, has been identified as a probable cause of a new syndrome of RSW in AD.

## 8. Implications for Developing an Inhibitor to HPRWSP

The goal for treating volume-depleted patients with RSW is to return them to a euvolemic state. The intuitive solution is to provide sufficient salt and water to improve the quality of life by establishing a euvolemic state at the expense of increasing urine output, especially sleep-depriving nocturia. This may be tolerable in patients whose RSW appears to subside after the successful treatment of their coexisting comorbid condition, although there are few who require salt and water supplementation for longer periods of time. This does not appear to be applicable to patients with AD, who seem to have a more permanent form of RSW. Developing an inhibitor would most effectively treat all RSW patients, especially those with a more persistent form of RSW as in AD.

The difference in duration of the salt wasting syndrome between those encountered in the hospital and AD raises interesting questions about the factors that control the upregulation of genes that control the production of HPRWSP. It appears that HPRWSP is normally produced at low rates with minimal or no clinical effects until there is an intercurrent illness that upregulates the gene that produces HPRWSP. Under normal conditions, therefore, the production of HPRWSP can be equated to a leaky faucet, which opens up for variable periods of time to induce RSW and reverts back to a less leaky state with the resolution of the comorbid condition. In AD, however, the faucet appears to be indefinitely open without an endpoint, suggesting that there is not only a prolonged or permanent upregulation of the gene but factors that also influence the rate of production of the putative factor. A permanently open faucet will, thus, indefinitely expose all organs of the body to the putative factor with no foreseeable endpoint.

Future investigations should include identification of the factors that affect the upregulation of the gene, production rates of HPRWSP and long-term exposure of all organs of the body to HPRWSP, especially the brain. Treatment of RSW in AD, therefore, introduces different challenges that require the development of an inhibitor of HPRWSP or that genetically downregulate its production once the diagnosis has been established instead of trying to create a euvolemic state by increasing salt and water intake and reducing the quality of life by the polyuria and nocturia. Inhibiting HPRWSP or reducing production of HPRWSP would have the greater benefit to the patient by eliminating any effects HPRWSP might have in other organs of the body, especially the dementia in AD. It appears that patients with an MMSE score less than 12 become progressively more volume depleted as their dementia worsens (Figure 5) [33]. Future studies must include patients at different stages of their disease to determine the onset of RSW and to determine if blood levels of HPRWSP will serve as a biomarker to identify the onset of RSW in AD.

Haptoglobin-related protein (HPR) is a relatively unknown primate-specific 45 kDa plasma protein that is produced by the liver and has a 91% sequence homology to haptoglobin phenotype Hp1-1 [39]. The newly synthesized protein contains hydrophobic short signal peptides with 16–30 amino acids usually located at the N terminus. The major function of fat-soluble signal peptides is to translocate the newly synthesized protein by attaching to transmembrane receptors and being secreted into the extracellular space with the signal peptide or without the signal peptide after cleavage by signal peptidases [40]. Although HPR is usually secreted without the signal peptide, the presence or absence of the signal peptide can determine ultimate function of the protein. It is uncertain what form of HPR contributes to its role as a high-affinity hemoglobin binding protein but it is clear that HPR with the signal peptide has no natriuretic activity and requires HPR to be without the signal peptide to attain a robust dose-dependent natriuresis [6]. However, HPR requires an attached signal peptide to combine to the high-density apoliproprotein L-1 (apoL-1) to form the trypanosome lytic factor (TLF-1), which requires the signal peptide to attach to the receptor on the trypanosome for endocytosis and eventual osmotic lysis of the trypanosome [41,42]. The ultimate utilization of HPRWSP as a biomarker of RSW in different populations requires assurances that HPR in plasma is purely without the signal peptide or with variable amounts of attached signal peptide. Although a western blot of the protein will identify the relative frequencies of HPR with and without the signal peptide, other simpler methods such as an ELISA would be in order.

## 9. Conclusions

We described the derivation and clinical utility of a superior approach to hyponatremia and hyponatremia-related diseases that should replace the archaic and ineffective volume approach. The intricate nuances exposed by this labor-intensive pathophysiologic approach has improved our ability to identify many of the causes of hyponatremia, the most important being the differentiation of SIADH from RSW and the high prevalence of RSW. The presence of natriuretic activity in the plasma of neurosurgical diseases and AD led eventually to identification of HPRWSP as the natriuretic factor in both groups of patients. Because haptoglobin-related protein with the signal peptide had no natriuretic activity, there is a need to determine whether HPR is present only without the signal peptide or with variable levels of the signal peptide. There is a need to simplify methods to determine HPRWSP levels in plasma that will serve as a much-needed biomarker for RSW in hyponatremia and hyponatremia-related diseases and to apply the appropriate therapy that will improve clinical outcomes. HPRWSP has multiple clinical applications that have been recently reviewed [43].

## Figures and Tables

**Figure 1 biomolecules-13-00638-f001:**
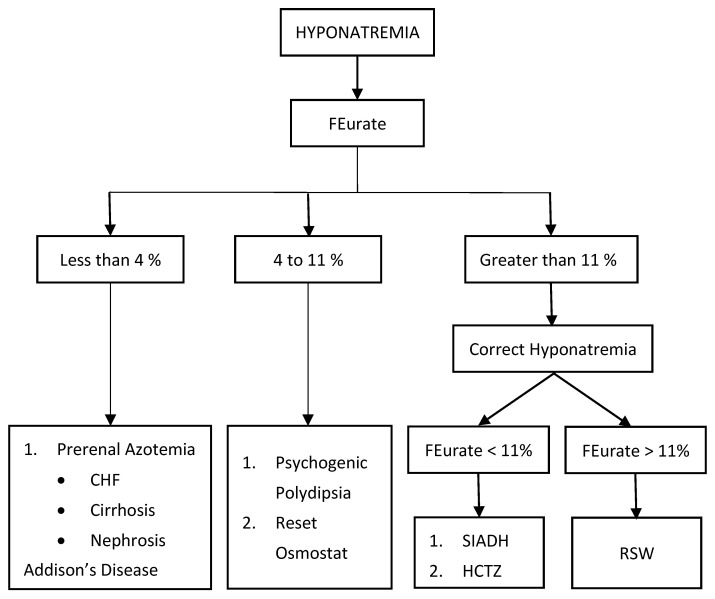
Algorithm showing the different causes of hyponatremia based on whether they have low- (<4%), normal- (4–11%) and high- (>11%) FEurate and whether infusion of isotonic saline infusions induce excretion of dilute urines or not in with Feurate >11% to differentiate SIADH from RSW. Because of a concern for the possibility of desalination occurring in patients receiving isotonic saline infusions, we suggest close monitoring of serum sodium when UNa + urine potassium concentration is >150 mmol/L.

**Figure 2 biomolecules-13-00638-f002:**
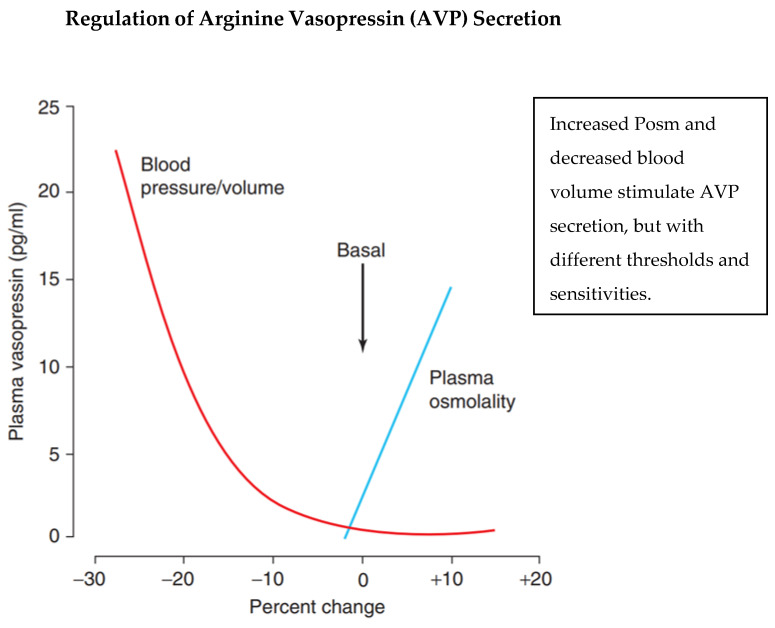
Graph showing the influence of plasma osmolality and extracellular volume on AVP/ADH levels in plasma. Note that the volume stimulus is more potent than the osmolar stimulus, so infusion of isotonic saline to a volume-depleted hyponatremic patient will eliminate the volume stimulus of ADH secretion and permit the hypo-osmolality to inhibit ADH secretion. Reprinted/adapted with permission from [19].

**Figure 3 biomolecules-13-00638-f003:**
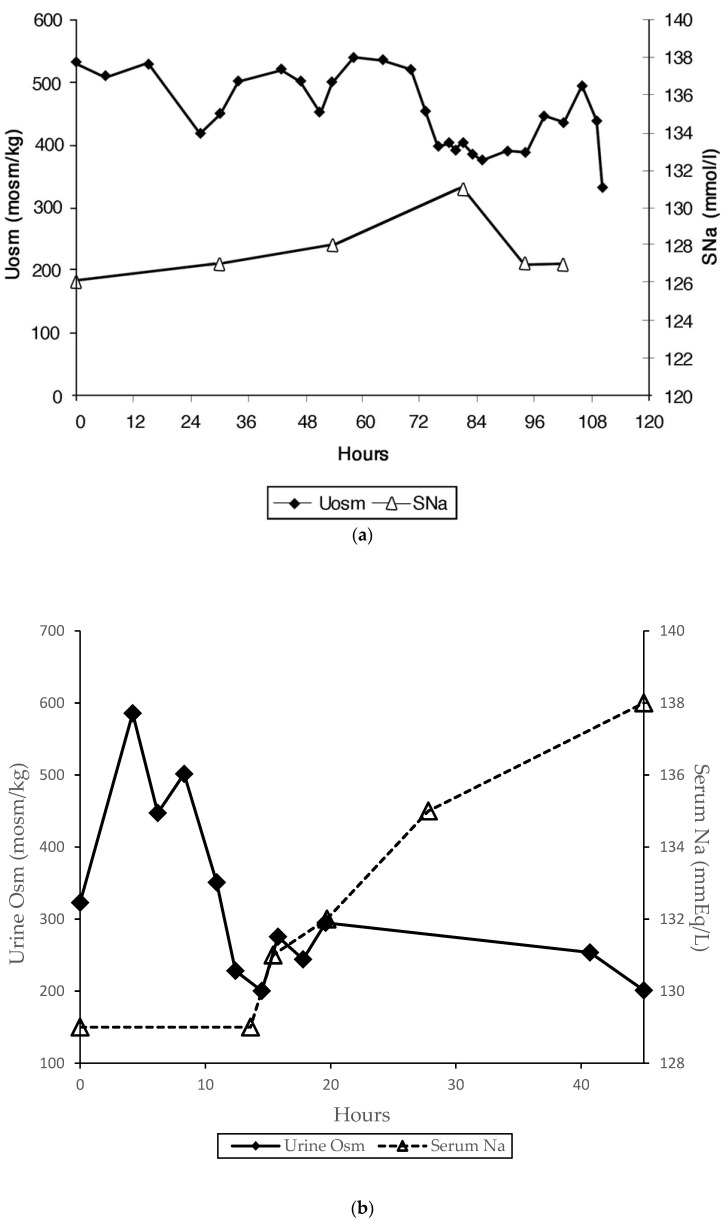
(**a**) Effect of isotonic infusions on urine osmolality and serum sodium concentrations in a patient with unequivocal SIADH based on an increase in blood volume determined by radio iodinated serum albumin and 51 chromium-labeled red blood cells, decreased plasma renin and aldosterone levels. Note the failure of isotonic saline to dilute the urine or correct the hyponatremia. Graph taken from reference [2]. (**b**) Effect of isotonic saline infusions on urine osmolality and serum sodium concentrations in a patient with unequivocal RSW based on a decreased blood volume by radio iodinated serum albumin and 51 chromium-labeled red blood cells, increase in plasma renin, aldosterone and ADH levels at baseline. Note the progressive decrease in urine osmolality after initiation of isotonic saline and eventual normalization of serum sodium within 48 h. Plasma ADH was undetectable when the urine osmolality was 152 mosm/kg. Reprinted/adapted with permission from [6].

**Figure 4 biomolecules-13-00638-f004:**
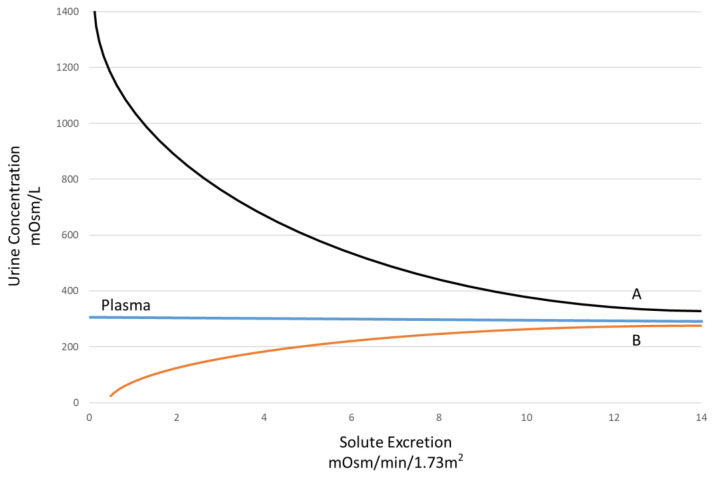
The effect of increasing solute excretion on urine osmolality under conditions of maximum and absence of ADH plasma levels. A is a study of subjects who had maximum ADH levels after a period of not drinking any water overnight and infusing hypertonic saline to maintain the high levels of ADH. B is a study of subjects who we given large volumes of water to generate maximally low urine osmolality when ADH levels were maximally suppressed. They were then infused with large volumes of hypotonic saline to maintain minimal ADH levels. The horizontal line represents plasma osmolality. Note how urine osmolality under maximum and minimum ADH levels in plasma gradually decreases or increases, respectively, to approach plasma osmolality when there is increasing solute excretion or increasing osmolar clearance. Graph taken from [21].

**Figure 5 biomolecules-13-00638-f005:**
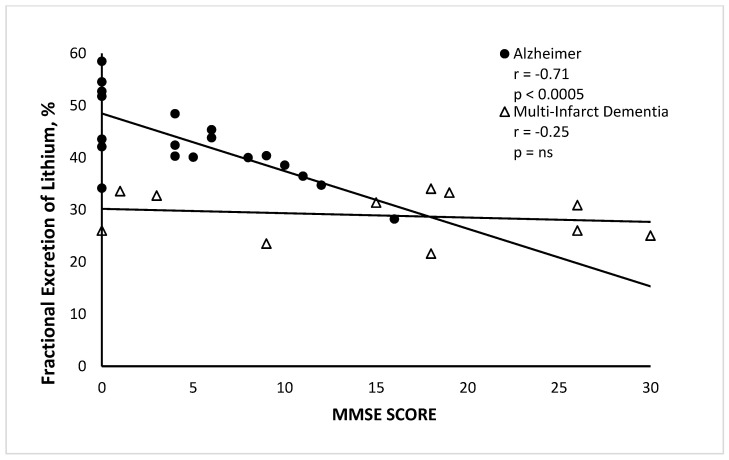
Graph demonstrating FElithium excretion rates in rats infused with plasma from patients with Alzheimer’s disease and multi-infarct dementia at different mini mental state examination scores.

**Figure 6 biomolecules-13-00638-f006:**
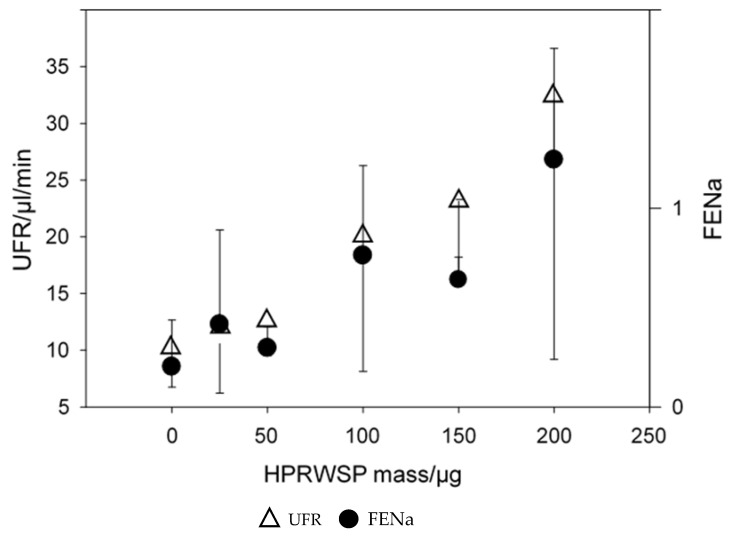
Effect of increasing dose of haptoglobin-related protein without signal peptide on FENa and urine flow rate. Dose response graph showing how increasing the dose of haptoglobin-related protein without signal peptide progressively increased fractional excretion of sodium and urine flow rates in rats.

**Table 1 biomolecules-13-00638-t001:** Table showing identical clinical characteristics of SIADH and RSW; the only difference is the volume status, being increased in SIADH and decreased in RSW.

Clinical and Laboratory Findings Common to SIADH and RSW.
Association with intracranial diseases
Hyponatremia
Concentrated urine
Urinary [Na] usually >30 mEq/L
Normal renal/adrenal/thyroid function
Non-edematous
Hypouricemia, Increased fractional excretion of urate
Only difference is volume status

## Data Availability

This is a review manuscript that provides no data other than those found exclusively in the literature where all such concerns would be applicable.

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
