# Peer review of "Haptoglobin-Related Protein without Signal Peptide as Biomarker of Renal Salt Wasting in Hyponatremia, Hyponatremia-Related Diseases and as New Syndrome in Alzheimer’s Disease"

_biomolecules, 2023, doi:10.3390/biom13040638_

Round 1

Reviewer 1 Report

General Comment: This is an interesting and thought provoking review paper that advocates for a new approach to hyponatremia.

Major Comments:

1. The review relies primarily on the authors' previously published papers. Is there an opposing point of view? If yes, then it should also be included for balance. Otherwise, this manuscript would be more appropriately labelled as a perspective rather than a review.

2. The figure legends do not cite the source of what is in the figure. Figures 2, 4, and 5 appear to contain data from a previous paper by the authors. This is clear in the text but the citation should also be in the figure legend. Figure 3 appears to be from reference 21 by a different author. If so, this citation should be in the figure legend.

3. The algorithm in table 2 shows administering isotonic saline to distinguish SIADH from RSW. Conventional practice is to avoid isotonic saline if one suspects SIADH, as it will worsen the hyponatremia. Is it safe to give isotonic saline outside of a clinical study? If so, how much is safe before causing harm? The authors should clarify this point so that clinicians do not cause harm.

4. Is the assay for HPRWSP available in hospital labs?

Minor Comments:

1. Lines 323-324 - ANP has also been shown to have effects in the collecting duct

2. There are typos in lines 96, 156, 182, 312, 313, 358, 369, 426, and 439

Reviewer 2 Report

I think that the authors did a great research work about hyponatremia and that the review summarizes their efforts and previous pubblications in a comprehensive way. Hyponatremia, particularly in the context of SIADH and RSW, is a difficult argument to deal with and is challenging in clinical practice. However, I think that some improvements are needed to make the review easily understandable.

Introduction:

- lines 64-73 “the identification … this important change in nomenclature”: as it is the introduction, it would be better to address the studies in more general terms, for example: “some studies conducted in the general medical ward increased our awareness that RSW might be more common than perceived including also patients without clinical evidence of cerebral disease ….”

- it would be useful to conclude the introduction with a statement that summarized the structure or the review in order to guide the reader, something like this: “in this review we firstly address the new pathophysiologic approach to …, then we summarize clinical studies … and finally describe the development and the applications of a new biomarker …”.

Rat clearance studies: this paragraph explains the experiments conducted in rats to demonstrate the presence of a natriuretic factor, however the second part, between line 261 and line 274, needs to be elucidate so it could be clear also for readers that did not read the original article: how was lithium administered? The authors chose to study lithium transport instead of sodium transport, but then they report both FElithium and FEsodium: please explain

lines 317-330: in this paragraph the role of ANP is discussed, please try to link better this paragraph with the previous one because ANP was not cited before it.

lines 361-377: please clarify if AD patients are normonatriemic because it is not explained

Figure 4 and figure 5 are exchanged: please amend their order and/or references in the text

Minor revisions:

in the abstract: line 32 HPRWSP was not cited before, please put the abbreviation in line 29

line 358 SAH: please amend

line 369: capital letter at the beginning of the sentence is needed

line 381: “establiching”: please amend

Figure 2: line 156: “not eh”: please amend

Figure 5 caption: line 312 “iwht” and line 313 “worens” please amend

Round 2

Reviewer 1 Report

The authors have responded satisfactorily to previous comments.